# High throughput transcriptomics analysis of ovine mammary epithelial cells stimulated with *Staphylococcus aureus in vitro*

**Saif Adil Abbood Al-Janabi**[1,2], **Ghulam Asghar Sajid**[1¤], **Sidra Zeb**[1],
**Muhammad Jasim Uddin**[3], **Mehmet Ulas Cinar** [1,4,5]*

1 Department of Animal Science, Faculty of Agriculture, Erciyes University, Kayseri, Türkiye, 2 Ministry of Agriculture, Office of Animal Resources, Baghdad, Iraq, 3 School of Veterinary Medicine, Murdoch University, Murdoch, Western Australia, Australia, 4 Department of Veterinary Microbiology and Pathology, College of Veterinary Medicine, Washington State University, Pullman, Washington, United States of America, 5 Betul Ziya Eren Genome and Stem Cell Center, Erciyes University, Kayseri, Türkiye

¤ Current address: Department of Infectious Diseases and Public Health, Jockey Club College of Veterinary Medicine and Life Sciences, City University of Hong Kong, Kowloon, Hong Kong
* mucinar@erciyes.edu.tr

## Abstract

In sheep, the innate immune response of mammary epithelial cells (MECs) plays a central role in combating mastitis, yet our understanding of their resistance mechanisms remains limited. This study aimed to elucidate the gene expression profiles of ovine MECs following in vitro stimulation with *Staphylococcus aureus* (*S. aureus*) using RNA-Seq technology. Bioinformatics analysis identified a total of 175 differentially expressed genes (DEGs), including 172 up-regulated and 3 down-regulated genes in the stimulated group compared to the non-stimulated control group. Gene ontology annotation and functional pathway analysis indicated that these DEGs are primarily involved in ribosomal functions, which are essential for protein synthesis and first target of pathogens, as well as in immune response dysregulations, infection, phagocytosis, and bacterial invasion of epithelial cells. Validation via quantitative real-time PCR (qRT-PCR) confirmed the RNA-Seq results. Our results revealed that DEGs converged on innate immune pathways (TLR, NOD-like receptor, *NF-κB*, MAPK), cytoskeletal remodeling and translational control, indicating inflammatory activation and cell injury in oMECs and highlighting candidate targets for mastitis resistance selection against *S. aureus*. These findings significantly contribute to the understanding of how ovine MECs respond to *S. aureus* stimulation, providing a foundation for further research, particularly regarding the immune defense mechanisms, strategies and implications in dairy industry.

**Data availability statement:** The data that support the findings of this study are available in the Gene Expression Omnibus (GEO) [https://www.ncbi.nlm.nih.gov/geo/]. Accession number: GSE295455.

**Funding:** This project was supported by Erciyes University Scientific Research Projects Unit under the code of FDK-2021-11437 and FYL-2023-12822.

**Competing interests:** The authors have declared that no competing interests exist.

## Introduction

Mastitis is a common infectious disease of mammary gland in dairy animals worldwide. It principally affects dairy cattle, it can also affect other milking producing ruminants, such as sheep and goats [1–3]. In these animals, mastitis is identified by inflammatory changes in the mammary glands (udder), leading to several adverse effects on health and production, such as reduced milk yield and compromised milk quality. Moreover, it results in economic losses for dairy farmers due to high veterinary expenses, and potential culling of the affected animals. Particularly in ewe production, it affects the normal growth, development, and survival of suckling lambs [4]. Mastitis can also lead to the death of ewes in severe cases [5]. Additionally, it raises welfare concern as it causes discomfort, anxiety, restlessness, feeding behavior alterations and pain in the diseased animals [6,7]. Mastitis is also important due to public health concerns, especially antimicrobial *S. aureus* strains [8].

Mastitis is mainly caused by microorganisms such as bacteria, viruses, or fungus, all of which can result in the development of mastitis [3]. A diverse staphylococcal species has been reported in mastitis cases, with varying prevalence with resistant genes such as *mecA*, which contribute to their persistence within the host population [9]. *Staphylococcus aureus* (*S. aureus*) is the most frequently involved and a major pathogen in clinical and subclinical mastitis infection [10,11]. Intramammary infections are commonly described as mild, chronic, or persistent [12]. Mammary epithelial cells (MEC) are specialized cells that form the lining of the mammary ducts and alveoli in the mammary tissue [5]. These cells not only play a crucial role in the production and secretion of milk in ruminants but also are an integral part of the defense mechanism in response to pathogen invasion [13]. However, during mastitis, it has been observed that *S. aureus* exhibits intracellular localization within MECs [14]. Upon the invasion of pathogens, MECs initiated an inflammatory response to combat and survival of intracellular activity. They release signaling molecules that attract immune cells to the infection site, contribute to the production of antimicrobial secretions, and engage in tissue repair processes [15]. Hence, it is important to understand the changing in the molecular activities after the invasion of *S. aureus* in ovine mammary epithelial cells (oMECs) [16,17]. RNA sequencing (RNA-Seq) provides potent tools to uncover the molecular mechanisms underlying development, differentiation, and disease [18,19]. The downstream differential expression analysis identifies the genes that are upregulated or downregulated in response to the infection, providing valuable insights into the immune response and molecular mechanisms underlying mastitis [20]. Functional pathways, gene ontology terms, or other relevant categories of these DEGs insight into the biological processes affected by the gene expression changes [21,22].

In recent years, many studies have been conducted to explore the gene expression patterns and molecular mechanisms associated with mastitis disease in cattle [23,24]. A significant alternation in gene expression was observed when bovine mammary tissue or epithelial cells were exposed to *S. aureus*. After in vitro infection, the pathological changes that occur in mammary gland are complex, while the molecular mechanisms involved in these changes remain unclear to make control and

treatment policies [17,25]. In sheep mastitis research primarily concentrated on its etiology and epizootiology [26], diagnosis, control through management, and treatment [27]. Worldwide, limited research has been conducted on the mammary gland transcriptome affected by sheep mastitis and its molecular pathogenesis using RNA-Seq [28,29]. In this study, our primary objective was to delve into the transcriptional regulation of oMECs following invasion by *S. aureus*. Through this exploration, we aimed to identify candidate transcripts involved in the immune response, with the aim of improving our understanding of the immune mechanisms triggered by *S. aureus*. By unraveling the molecular complexity of the host response, this study provides valuable information that could pave the way for further research and more effective strategies in both animal breeding and therapeutic interventions.

## Materials and methods

### Animal selection and sampling

Experimental procedures used in this study were performed according to the Institutional Animal Care and Use Committee of Erciyes University, Kayseri, Türkiye, and the research protocol adhered to the Turkish Council on Animal Experiment guidelines on farm animal facilities (15 February 2014, #28914). Any pre-existing mammary infection can affect gene expression; therefore, samples were collected from physiologically healthy ewes. Healthy ewes of Akkaraman breed were selected ($n=3$) based on standard physical examinations with an age ranging 2–3 years. The possible lesions on the skin of the udder have been checked for any abnormality in the shape of udder (increase in size and atrophy) before slaughtering. Then, the mammary glands were palpated, including the teats (shape, size, temperature, and consistency) were checked for each quarter. Also, any pain reaction of the animal and any swelling/heat in lymph nodes. Animals were slaughtered in a commercial abattoir in Kayseri province. Tissues were collected aseptically from the mammary parenchyma immediately after slaughter of selected ewes. Samples were then transferred to the laboratory in chilled 1×Dulbecco phosphate-buffered saline (DPBS, without calcium and magnesium, Sigma-Aldrich, USA) for cell isolation and transformed primary cell culture.

### Ovine mammary epithelial cell culture

Ovine mammary epithelial cell culture was established in 40 mL of Dulbecco's modified Eagle's medium (DMEM, 500 mL, high glucose, Sigma-Aldrich, USA) with the addition of 20% bovine serum (FBS, 10 mL, Sigma-Aldrich, USA), insulin (5 μL, Sigma-Aldrich, USA), amphotericin B (500 μL, Sigma-Aldrich, USA) and penicillin-streptomycin (250 μL, Sigma-Aldrich, USA). The cells were cultured on uncoated polystyrene Petri dishes with surface modifications designed to enhance cell culture (35 mm). Cultures were maintained in a 5% $CO_2$ atmosphere at 37 °C until monolayer confluence was achieved. Cell identification was performed using microscopy, and the cell surface marker *EpCAM* [30] was assessed by quantitative real-time polymerase chain reaction (qRT-PCR) expression analysis keeping *GAPDH* as reference gene (S1 Fig) as used by Al-Janabi et al. 2023 [31].

### *S. aureus* culture and identification

Milk samples from ewes with confirmed mastitis field cases were plated on blood agar plates (Oxoid, UK) supplemented with 5% defibrinated ovine blood and incubated at 37 °C for 24–48 hours. Following incubation, suspected bacterial colonies were subjected to Gram staining and examined under a light microscope. Gram-positive cocci were further characterized using a series of biochemical tests. The biochemical properties of the isolates were assessed through catalase activity, hemolysis on blood agar, coagulase activity, nitrate reduction, DNase agar, clumping factor presence, arginine dihydrolase activity, and urease production, as described by Quinn et al. (1998) [32]. Coagulase-positive isolates were further analyzed using the API STAPH IDENT system, 32 Staph (bioMérieux SA, 69280 Marcy-l'Étoile, France), to confirm the identification of *S. aureus*.

## Stimulation model

A field strain of *S. aureus* was isolated at the Microbiology Laboratory of the Veterinary Faculty at Erciyes University, from a diseased animal with known mastitis pathogenesis. A fresh microbial culture was prepared in Tryptic Soy Broth (TSB) by overnight incubation and was subsequently washed twice with PBS before the planned treatment. The microbial stimulation solution was prepared in Dulbecco's modified Eagle's medium (DMEM, 500 mL, high glucose, Sigma-Aldrich, USA) without antibiotics. A concentration of $1.5 \times 10^8$ microbial cells/mL was obtained by using the 1 OD at 620 nm. Multiplicity of infection (MOI) was determined using our early pilot studies, 600:1 MOI was used to challenge oMECs in wells containing 2 mL of culture (up to $2.5 \times 10^5$ cells/well). Control well with the same quantity of cells and volume of Dulbecco's modified Eagle's medium (DMEM, 500 mL, high glucose, Sigma-Aldrich, USA) were incubated at 37 °C for 24 hours, along with treated wells.

## Total RNA isolation and cDNA library preparation

Total RNA isolation was performed from the control and treatment groups (three samples per group) using the Trizol isolation kit (TransZol Up Plus, ER501-01-V2, China) according to the manufacturer's instructions. The quality and quantity of RNA was assessed using PacBio Nano drop spectrophotometer with absorbance measurement at 260 nm, and A260/A280 ratio that ranged from 1.8 to 2.2, respectively. RNA samples from three control and three treatment groups were used for high throughput sequencing. cDNA libraries were constructed from the total RNA by using the TruSeq RNA Library Preparation Kit (Illumina, CA, USA) according to the manufacturer's protocol, then sequencing was done on the Illumina HiSeq 4000 platform to obtain paired end reads.

## Quality control of raw data

A systemic approach was adopted to prepare the original raw data for performing a standardized bioinformatics analysis. Initial quality assessment was performed to check the status of raw data. Clean data obtained after trimming the low-quality reads (sequences with errors or low base call quality), overlapping adapters, ribosomal RNA and PCR duplicates using HTStream (https://github.com/s4hts/HTStream). The quality of clean reads was evaluated using FastQC.

## Read alignment and differential expression analysis

Clean reads of each sample were used to generate the index file of the sheep reference genome (https://www.ncbi.nlm.nih.gov/datasets/genome/GCF_016772045.1), and alignment of the paired end clean reads to the assembly was performed. Salmon was used to obtain the count reads of each sample. The DESeq2 package in R identified the differential expressed genes (DEGs) between the control and *S. aureus* stimulation group. DESeq2 applies a shrinkage estimator for dispersion estimates to improve the stability of results. Dispersion estimates for each gene were shrunk toward a fitted trend using an empirical Bayes approach, ensuring that estimates for genes with low counts were more reliable. This approach reduces overestimation of dispersion in lowly expressed genes, leading to improved model fitting and a lower false positive rate. It is determined that the DEGs with p value < 0.05 and | log2 (fold change) | ≥ 1 are regarded as thresholds significant differential expression in stimulation group as compared to control group.

## GO annotation and pathway enrichment analysis

GeneXplain tool (https://genexplain.com) was used to perform GO annotation while pathway enrichment analysis was carried out using ConsensusPathDB (http://cpdb.molgen.mpg.de) to gain a holistic understanding the role of the DEGs in mammary epithelial cells on the basis of biological system functions. GO terms with p value < 0.05 were considered significant enrichment. Pathway enrichment data was visualized using ggplot2 in R.

### Protein-protein interaction

Understanding the functional interactions among expressed proteins is important for a thorough grasp of cellular molecular functions. The ConsensusPathDB database was used to integrate known and predicted protein-protein association data from the DEGs of current study. Network analysis of protein-protein interactions associated with DEGs was carried out using ConsensusPathDB [33] and the standard default visualization was used to represent his PPI network.

### Quantitative real-time PCR (qRT-PCR) validation of DEGs

Validation of DEGs was performed on the same samples (used in RNA sequencing) of control and *S. aureus* stimulation group using qRT-PCR technique. The cDNA was prepared by reverse transcriptase reaction according to manufacturer's instructions. The primers were designed using online software Primer 3 (https://primer3.ut.ee) and presented in S1 Table. The *GAPDH* gene was used as endogenous control. The qRT-PCR was carried out using the SYBR Green PCR Master Mix kit (Bio-Rad, CA, USA) with the Light Cycler 96 instrument (Roche, Basel, Switzerland) according to the manufacturer instructions.

### Experimental design and statistical analysis

In this reliability experiment, the same total RNA samples were used as those that were utilized for RNA-Sequencing. Six DEGs, comprising five upregulated and one downregulated gene were selected for validation. Each test was conducted on six samples (three control and three stimulated) and repeated three times for each sample. Data analysis and comparison were carried out using Microsoft Excel 365 (Microsoft Corporation, Washington, USA). Relative mRNA levels were determined using a comparative ct ($2^{-\Delta\Delta CT}$) method as done earlier [34] and compared with RNA-Seq results. Graph visualization was generated using ggplot2 3.4.4 package on RStudio (version: 2023.12.0 + 369, RStudio, Boston, USA).

## Results

### Sequencing data statistics

In this study, a total of 6 cDNA libraries were constructed by isolating total RNA from *S. aureus* stimulated oMECs and control groups. The proportion of duplicate reads across all sequenced datasets ranged between 91% and 93% in paired end reads, indicating a high level of redundancy within the sequencing data. After trimming and removal of duplicates, clean data obtained for down-stream analysis (Table 1).

### Differentially expressed gene analysis in ovine MECs

Differential expression testing was carried out by shrinking estimator method. High proportion of genes were shrunk toward curve revealed fitting of the model and less chances of false positive results (S3 Fig). DESeq2 was employed to screen the differentially expressed genes among stimulation and control groups. For normalization, DESeq2's variance stabilizing transformation method was used to correct the differences in sequencing depth and variability. The statistics of

**Table 1. Summary of the RNA-Seq data quality.**

| Sample | Control-1 | | Control-2 | | Control-3 | | Stimulated-1 | | Stimulated-2 | | Stimulated-3 | |
|---|---|---|---|---|---|---|---|---|---|---|---|---|
| **Paired reads** | R1 | R2 | R1 | R2 | R1 | R2 | R1 | R2 | R1 | R2 | R1 | R2 |
| **Raw reads (Millions)** | 10.3 | 10.3 | 12.7 | 12.7 | 9.6 | 9.6 | 8.7 | 8.7 | 7.0 | 7.0 | 8.0 | 8.0 |
| **Clean reads (Millions)** | 0.5 | 0.5 | 0.5 | 0.5 | 0.5 | 0.5 | 0.5 | 0.5 | 0.4 | 0.4 | 0.4 | 0.4 |
| **Average read length (base pair)** | 147 | 146 | 146 | 146 | 146 | 146 | 147 | 147 | 147 | 147 | 147 | 147 |
| **GC%** | 50 | 50 | 50 | 50 | 47 | 47 | 50 | 50 | 50 | 50 | 50 | 50 |

number of DEGs were based on significance level p < 0.05 and | log2 (fold change) | ≥ 1 as shown in Fig 1. A total of 175 genes were identified as differentially expressed in the *S. aureus* stimulation group compared to control group, with 172 genes being up-regulated genes and only 3 genes being down-regulated (Fig 2). Among 175 DEGs enlisted in S2 Table, 40 were identified as novel unannotated genes.

## Functional analysis of DEGs

The pathway enrichment analysis of DEGs revealed that most of them are involved in production and activation of ribosomes and spliceosomes, and subsequently RNA, mRNA, and other nucleic acid bindings, indicating the activation of immune response. Upregulated genes such as *RPS23* and *RPL32* play a central role in ribosome activity and are associated with the pathogenesis of coronavirus infection. *ACTB* is involved in processes such as bacterial invasion of epithelial cells, apoptosis, and phagosome formation. Furthermore, *CYRIB* is involved in the regulation of memory T cell activation, and *PFN1* is involved in the Rap1 signaling pathway, regulation of the actin cytoskeleton, and response to salmonella infection. *UBC* and *HUWE1* are involved in ubiquitin-mediated protein degradation and contribute to mitophagy and maintenance of proteostasis. Genes such as *YBX1* and *DDX5* are involved in RNA binding and processing and play a role in C5 methylcytidine-containing RNA binding and spliceosome activity. The identified downregulated gene *NEFM* is associated with molecular functions and signaling pathways associated with neurodegeneration. Although the functions of novel DEGs such as *LOC114112704* and *LOC114112490* have not yet been determined. Results of top 25 DEGs pathway enrichment are presented in Fig 3. The Gene Ontology (GO) annotation was performed to explore the molecular functions of DEGs based on three categories molecular function (MF), biological process (BP) and cellular compartment (CC). Notably, ribosomes functions, RNA and nucleic acid binding, heterocyclic compound binding, translation regulator activity, as well as cadherin binding were significantly overrepresented (Fig 4, S4 and S5 Figs). PPI network shows the predicted protein interactions.

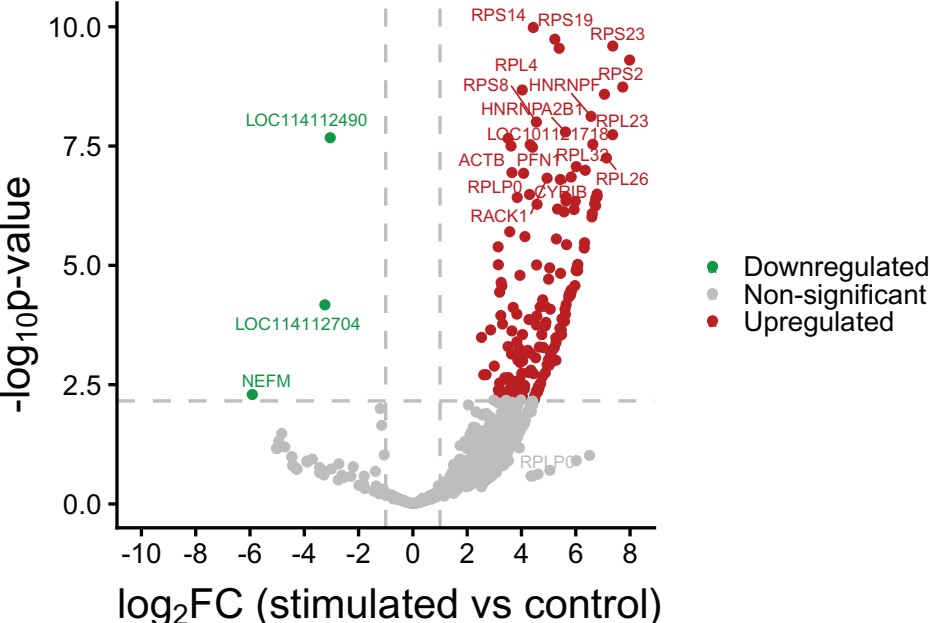

**Fig 1. Volcano plot of the statistically significant DEGs, represented by red (up-regulated) and green (down-regulated) dots (p ≤ 0.05 and | log2 (fold change) | ≥ 1) in *S. aureus* stimulated samples as compared to control samples.**

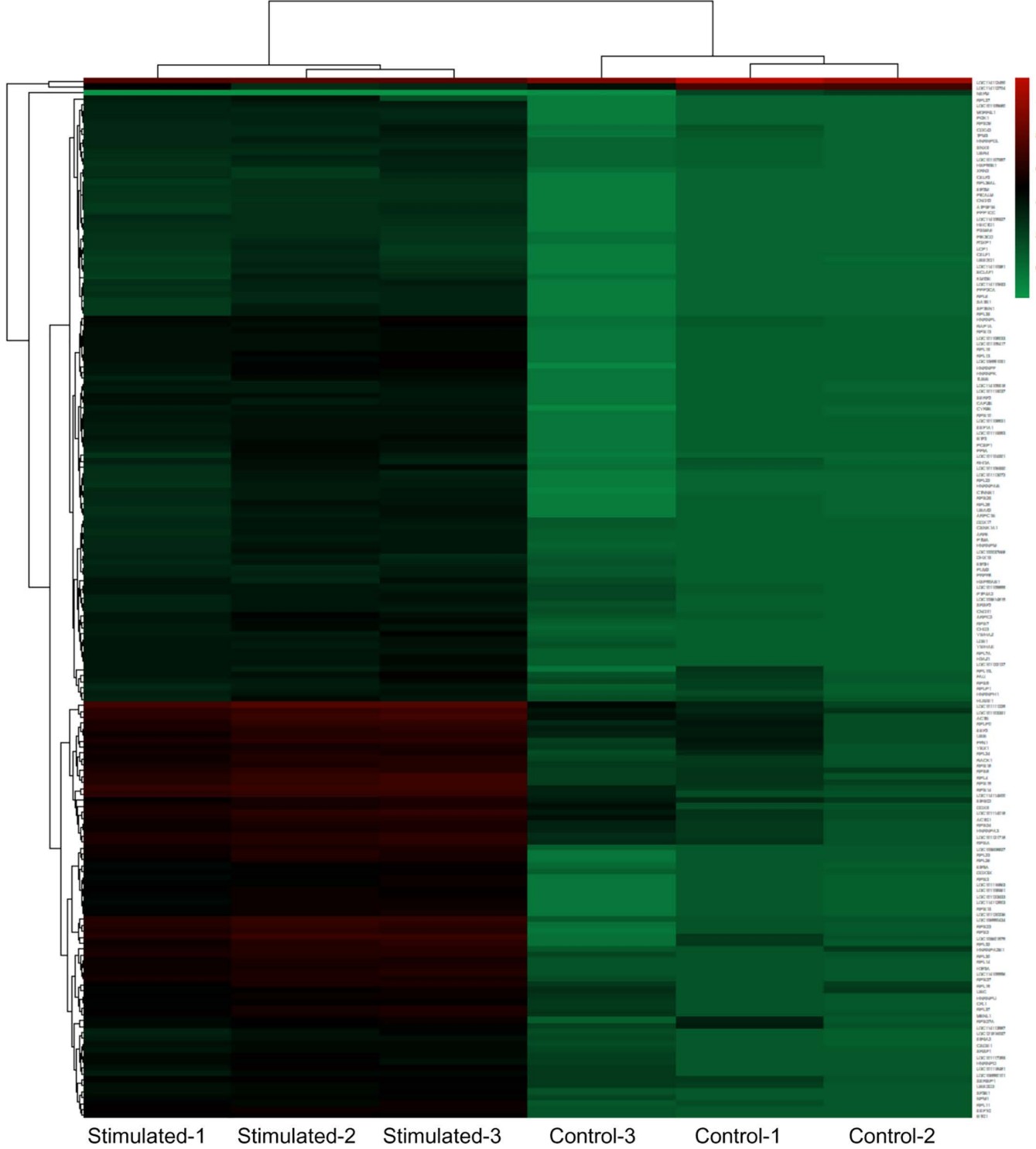

**Fig 2. Cluster analysis of significant differentially expressed genes in ovine mammary epithelial cells between the *S. aureus* treated samples (Stimulated-1, Stimulated-2, and Stimulated-3) and their control samples (Control-1, Control-2, and Control-3). Red color shows high expression level while green color represents low expression level of genes.**

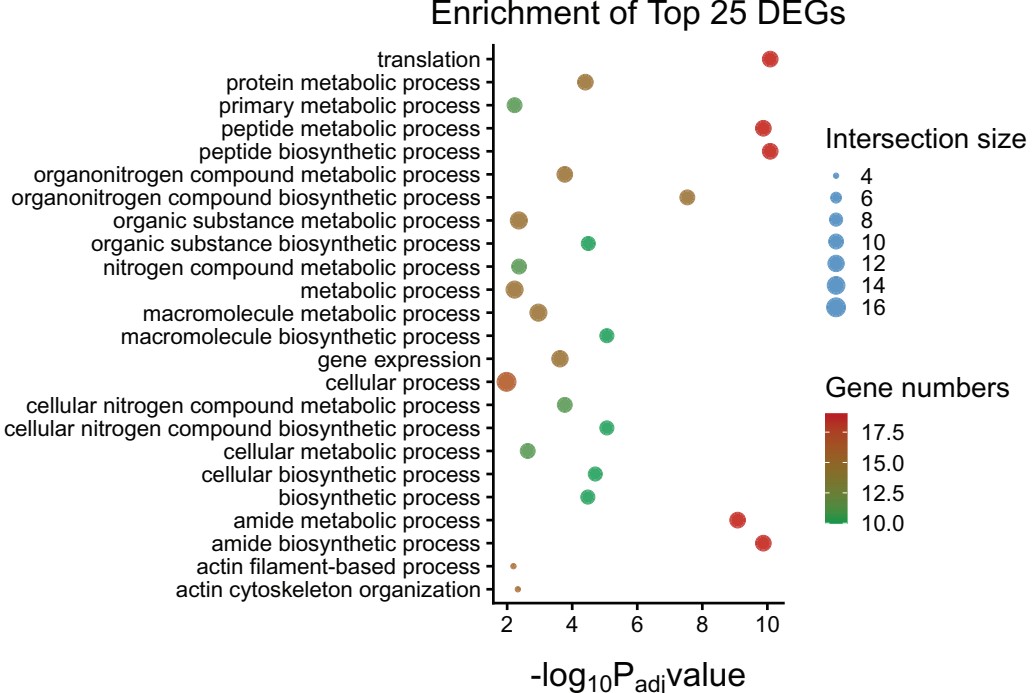

**Fig 3. Pathway enrichment of top 25 DEGs in ovine mammary epithelial cells.** Biological pathways displayed on left axis, and the size of the circle indicates the intersected genes while the color represents the numbers of DEGs involved in pathway.

## Protein-protein interaction (PPI) analysis of DEGs

To overview of functional relationship between DEGs, protein-protein interaction (PPI) analysis was performed. The resulting network, as illustrated in Fig 5, highlights key molecular complexes and their interactions. Prominent clusters included the ribosomal complex (both 40S and 60S subunits), Nop56p-associated pre-rRNA complex, and heterogeneous nuclear ribonucleoproteins (HNRP) proteins.

## Validation of DEGs by qRT-PCR

Validation of the accuracy of RNA-Seq results were performed by selecting (based on their relevance to immune response and statistical significance) 10 genes (9 up-regulated and 1 down-regulated) with qRT-PCR testing keeping *GAPDH* gene as internal control. The results showed that the relative expression of the selected genes was consistent with RNA-Seq results, indicating that the study was reliable (Fig 6).

## Discussion

Mastitis is a common disease in dairy animals, but its impact extends to meat flocks as reduced milk production in ewes can lead to suboptimal growth. The most prevalent microorganism in small ruminant intramammary infections is *Staphylococci*, and certain stains of *S. aureus* are responsible for clinical and sub clinical mastitis in ewes [35]. Studies have shown that *S. aureus* causes significant impact on the health of the udder by adhering to mammary epithelial cells, leading to tissue damage and self-protection from the host immune system [36,37]. It is important to deepen the understanding of this infection and response of the mammary tissue at an intra molecular level. The current study was designed to assess the response of oMECs using high-throughput sequencing of total RNA after stimulation with *S. aureus*.

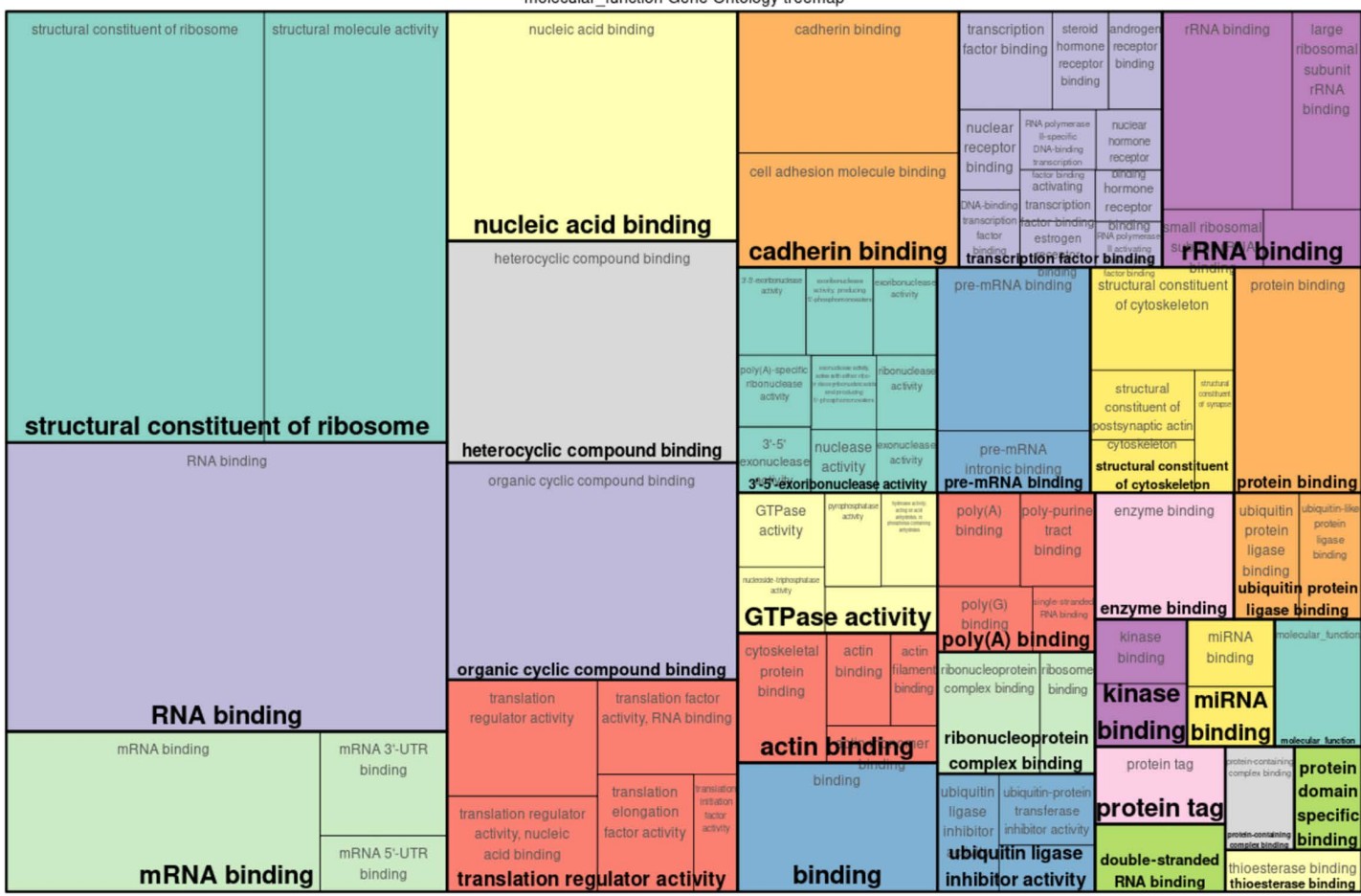

**Fig 4. Tree map illustrating GO annotation associated with DEGs.** The boxes are organized into clusters corresponding to the upper hierarchy GO-term which are highlighted in bold letters. Relative block size indicates the frequency of associated genes within categories.

Identification of DEGs, their pathway enrichment, and functional annotation analysis contributed to the better understanding of *S. aureus* infection and its immunological and biological implications, especially in sheep.

In current study the on average 0.5 million clean reads were mapped with the sheep genome while 38% of reads were not accomplished with the reference genome. This was probably due to lack of genome information availability for sheep. The high degree of duplicate reads (70–95%) in RNA-Seq data suggested by [38] can be attributed not only to PCR amplification but also to read mapping space saturation caused by real biology of high expression levels. The high duplicate read proportion in our dataset likely reflects the dominance of a small subset of highly expressed transcripts rather than sequencing artefacts, consistent with previous RNA-Seq reports, and does not undermine DEG reliability.

Here we screened 175 DEGs with p ≤ 0.05 and | log2 (fold change) | ≥ 1 in oMECs, 172 genes were up-regulated while 3 were down-regulated. Chen *et al*. [1] identified a total of 186 DEGs in bovine mammary epithelial cells after treatment with *S. aureus*, of which 31 were up-regulated while 155 DEGs were down-regulated. In another study, 259 DEGs were identified following the treatment of *S. aureus* on bovine mammary epithelial cells, with 124 DEGs displaying up-regulation while 135 were down-regulated [17]. A total of 194 DEGs were identified after an intra-mammary injection of *S. aureus* in cow, 154 were up-regulated and 40 genes were down-regulated [39]. Variations and distinctions in expression trends were

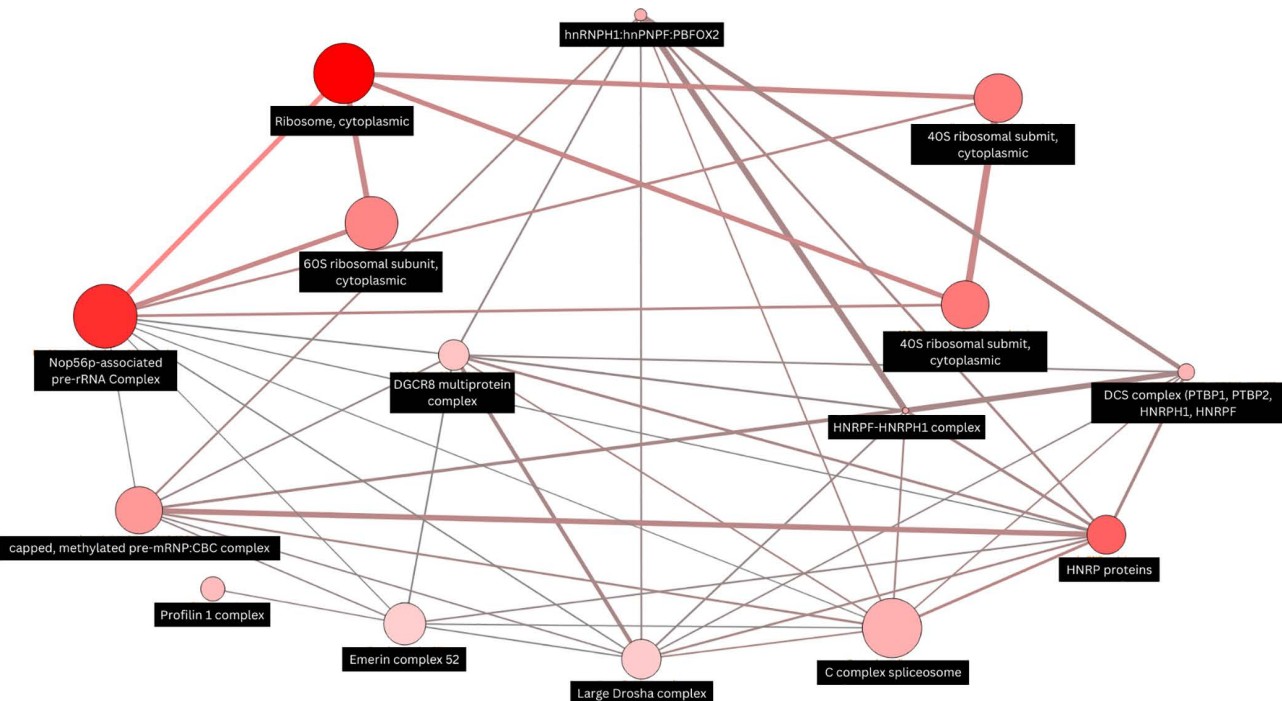

**Fig 5. Protein-protein interaction (PPI) analysis of DEGs.** Map node size and color to degree, low values to small sizes and dark colors. Map connecting line width indicates the strength of interactions.

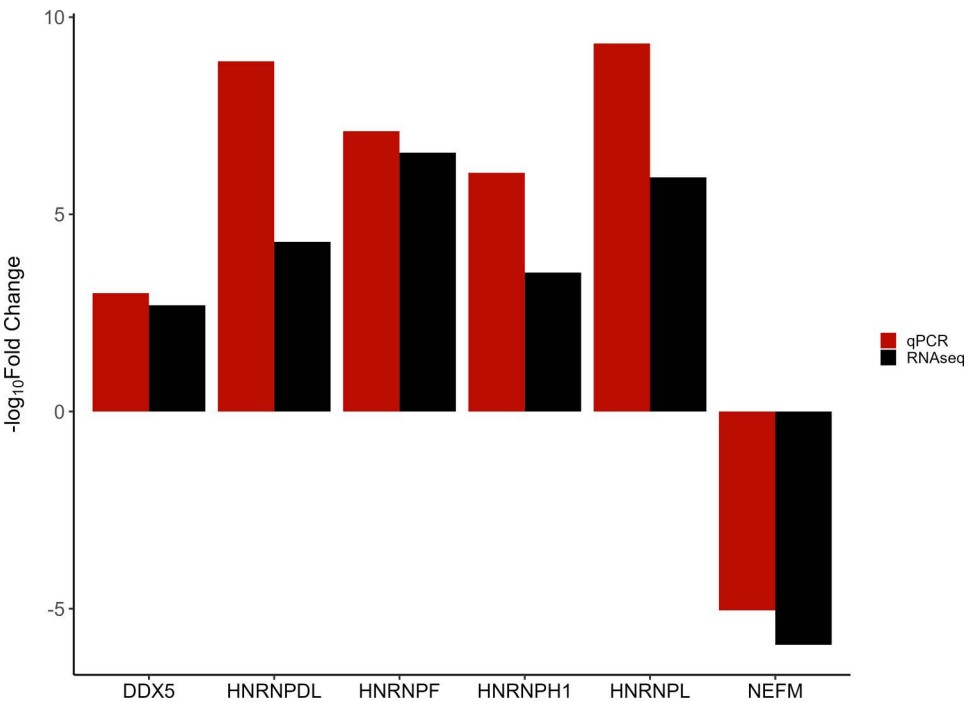

**Fig 6. mRNA expression levels of selected DEGs quantified by qRT-PCR.** *GAPDH* was used as internal control and data represented as -log10 of fold change (*n* = 3 samples per group).

noted from prior studies, and might be attributed to differences in experimental design, as well as variation in the genetic makeup of animal species and microbial strain.

Several DEGs of current study are involved in host-pathogen interactions and responding to foreign stimuli. Among top 25 DEGs, the upregulation of *RPS19* gene mediate cap-dependent translation [40], MIF, ERK and NF-κβ and interact with pathogen proteins [39] during the immune signaling. *RPS14* influences TLR-4 pathway in recognizing the *S. aureus* and initiating the immune system during the mammary gland infection [41–43]. Multiple members of the heterogeneous nuclear ribonucleoproteins (hnRNPs) family such as *hnRNPDL*, *hnRNPA0*, *hnRNPH1* and *hnRNPM* were up regulated in current study. This family is conserved for RNA-binding proteins that have a critical role in cellular processes, including transcription, post-transcriptional modification, and translation [44]. HnRNPs regulate innate as well as adaptive immunity in response to bacterial infections [45] such as HnPNPA0 binds to specific sequence of inflammatory genes including *TNF-α* and *IL-6*, controlling the inflammatory response [46]. Up regulation of *hnRNPM* after 4 hours of *Salmonella* infection stimulated the chemokine receptor *CCRL2*, the regulator of NF-κB (*NFKBIZ*) pathway [47]. Another up regulated gene *ACTB* have also role in cell migration, invasion, and dysregulation of cytoskeleton [48]. The gene ontology functional enrichment analysis also revealed that most of differentially expressed genes were involved in the molecular function of structural constituent and molecular activity of ribosomes, indicating that the protein synthesis machinery was disturbed in MECs after the stimulation with the *S. aureus*. The *S. aureus* invasion of MECs involves an active participation of cytoskeleton of mammary tissue and host translation response is well reported [49,50]. Cytoskeleton plays a significant role in all aspects of immune system function at all levels of infection, from the early immune cell development to the later stages of immune responses, including recruitment, migration, signaling, and activation of both innate and adaptive immune components [51]. The *S. aureus* modulated cytoskeleton of MECs during the internalization process using actin-dependent cytoskeleton pathway [52]. Taken together, these results suggested that there was unbalanced immune suppression along with the activation of inflammatory immune response and cell damage observed with the *in vitro* stimulation of oMECs with *S. aureus*, providing a deeper insight into the responsible mechanisms. These genes integrate into key innate immune pathways including TLR, NF-κB, NOD-like receptor, and MAPK signaling, thereby linking transcriptional changes to functional immune outcomes in oMECs. Furthermore, these transcriptomic shifts highlight immune targets that might be complemented by interventions with dual antibacterial and antioxidant activity, such as copper nanoparticles [53].

Beyond the immunological context, the transcriptomic data of the current study also provides insights into fundamental cellular and molecular mechanisms that underpin the response of oMECs to *S. aureus* stimulation. The regulation of several DEGs such as *ACTB*, *ACTG1*, and *CDC42* underscores the critical involvement of the cytoskeleton in maintaining cellular integrity and facilitating pathogen internalization, as well as subsequent intracellular trafficking [53,54]. Moreover, transcriptional changes in genes associated with protein synthesis and cellular metabolism, such as *EEF1A1* and *RPS14*, suggest a diversion of host cellular machinery to address the metabolic demands imposed by bacterial invasion [55]. These molecular adaptations mirror findings in similar transcriptomic studies on mammary epithelial cells in livestock [56], emphasizing conserved host cellular mechanisms during bacterial infections. These observations complement the immunological findings, thus presenting a comprehensive understanding of how *S. aureus* interacts with oMECs at both cellular and molecular levels.

## Conclusions

In this study, we evaluated the immune and cellular response of *S. aureus* stimulated oMECs through whole transcriptome profiling. When *S. aureus* invades ovine mammary epithelial cells, it triggers an immune response, activates transcriptional machinery, and induces the expression of genes related to immunity, diseases, and cell damage. The DEGs may be critical in understanding molecular mechanisms prevailing with the invasion of *S. aureus* in oMECs. This study provides novel insight that could lay a foundation for the screening of the genes related to mastitis resistance specific to *S. aureus* origin thus help in the selection of mastitis resistant animals.

## Supporting information

**S1 Fig. The identification and validation of ovine mammary epithelial cells before *S. aureus* treatment.** (a) Optical microscope image of cells at 200 μm scale (b) The expression of EpCAM gene, the surface cell marker of mammary epithelial cells.
(TIF)

**S2 Fig. PCA plot of control (red) and *S. aureus* stimulated (blue) group.**
(TIF)

**S3 Fig. Dispersion plot of shrink gene-wise dispersion estimates towards the GLM fitted line.**
(TIF)

**S4 Fig. Tree map illustrating biological processes of GO annotation associated with DEGs.** The boxes are organized into clusters corresponding to the upper hierarchy GO-term which are highlighted in bold letters.
(TIF)

**S5 Fig. Tree map illustrating biological processes of GO annotation associated with DEGs.** The boxes are organized into clusters corresponding to the upper hierarchy GO-term which are highlighted in bold letters.
(TIF)

**S1 Table. List of primers used for qRT-PCR.**
(DOCX)

**S2 Table. List of differentially expressed genes as compared to control (non-stimulated) ovine epithelial cells with *S. aureus in vitro.***
(DOCX)

## Acknowledgments

The authors highly appreciate Prof. Dr. Kadir Semih Gümüşsoy, Department of Veterinary Microbiology in the Faculty of Veterinary Medicine at Erciyes University, for providing the microbial culture of *S. aureus*. The authors are also indebted to Res. Asst. Mr. Mustafa Özdemir for his help during the experiment.

## Author contributions

**Conceptualization:** Mehmet Ulas Cinar.

**Data curation:** Ghulam Asghar Sajid.

**Formal analysis:** Saif Adil Abbood Al-Janabi.

**Funding acquisition:** Mehmet Ulas Cinar.

**Methodology:** Saif Adil Abbood Al-Janabi, Ghulam Asghar Sajid, Muhammad Jasim Uddin.

**Software:** Ghulam Asghar Sajid.

**Supervision:** Muhammad Jasim Uddin, Mehmet Ulas Cinar.

**Validation:** Sidra Zeb.

**Writing – original draft:** Saif Adil Abbood Al-Janabi, Ghulam Asghar Sajid, Sidra Zeb.

**Writing – review & editing:** Muhammad Jasim Uddin, Mehmet Ulas Cinar.

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
