## [Decision Letter · Decision Letter 0]

13 Mar 2025

Dear Dr. Cinar,

Thank you for submitting your manuscript to PLOS ONE. After careful consideration, we feel that it has merit but does not fully meet PLOS ONE’s publication criteria as it currently stands. Therefore, we invite you to submit a revised version of the manuscript that addresses the points raised during the review process.

We look forward to receiving your revised manuscript.

Kind regards,

Muhammad Ahmad

Academic Editor

PLOS ONE

Journal Requirements:

2. Thank you for including the following ethics statement on the submission details page:

'Experimental procedures used in this study were performed according to the Turkish council on animal experiment guidelines on farm animal facilities (15 February 2014, #28914).'

Please clarify the full name of the ethics committee that approved your animal study.

3. Thank you for stating the following financial disclosure: [This project was supported by Erciyes University Scientific Research Projects Unit under the code of FDK-2021-11437 and FYL-2023-12822.]. 

4. Please include captions for your Supporting Information files at the end of your manuscript, and update any in-text citations to match accordingly. Please see our Supporting Information guidelines for more information: http://journals.plos.org/plosone/s/supporting-information .

Reviewers' comments:

Reviewer's Responses to Questions

**Comments to the Author**

1. Is the manuscript technically sound, and do the data support the conclusions?

Reviewer #1: No

Reviewer #2: Yes

Reviewer #3: Yes

2. Has the statistical analysis been performed appropriately and rigorously?

Reviewer #1: No

Reviewer #2: Yes

Reviewer #3: Yes

3. Have the authors made all data underlying the findings in their manuscript fully available?

Reviewer #1: No

Reviewer #2: Yes

Reviewer #3: Yes

4. Is the manuscript presented in an intelligible fashion and written in standard English?

Reviewer #1: Yes

Reviewer #2: Yes

Reviewer #3: Yes

Reviewer #1: Dear authors,

understanding the complex mechanisms of immune response against pathogens is of particular importance. Your manuscript deals with the infection of ovine mammary epithelial cells with S. aureus.

The manuscript is nicely structured; however, a lot of information is missing to get a clear picture of the whole experiment. Further, some sections are described superficially, especially in the method and result part.

Line 87: More information is needed on the ewes which were slaughter. Please add information related to the type of breed, the age of the animals and the number of slaughtered animals.

Line 102/103: You collected samples from different ewes; where these samples pooled before establishing the cell culture model?

How many replications were performed in the cell culture experiment? I have to look into the results section to get this information. Please provide the number here as well.

Line 106: Please provide a reference for EpCAM.

Line 145 to 147: You describe the thresholds for genes to be significantly differentially expressed. The problem of multiple test (the importance of false discovery) is missing. In the result section (192 to 194) a method is described to control the false positive results. This method needs to be explained in the method section as well, and which type this method controls false positive. Please also provide further information on the applied method.

Line 164: Which samples have been used here?

As mention above, the result section is too superficial and needs to be rewritten. Figure 5 and S2 are not described in the text or referenced. The results are presented in tables or figures, but they are not described in the text. It is left unclear how to selected 6 genes for the validation step.

Line 186: “there was a higher percentage of duplicated ….” Compared to which group? Please be more specific.

Line 189: In table 1 you see differences between the raw reads of the control group and the stimulated group? Please give an explanation.

Line 227 to 230: Why were some GO groups overrepresented? How did you observe this from the given figures?

The discussion mainly focused on the functional description of the genes, because there is a lack of comparable studies. An outlook on further research is missing. The listed experiments were performed in other species. Could you provide additional information on the experiment design in order to classify the obtained results here?

Reviewer #2: In the current manuscript, authors conducted a transcriptomic study to identify host defense genes in ovine mammary epithelial cells challenged with a field strain of Staphylococcus aureus. Overall, the study and manuscript appear satisfactory, though I have a few minor concerns.

1. Inclusion of Gene Abbreviations: It would be beneficial for the authors to include the full names or abbreviations for all the genes mentioned throughout the manuscript. This would improve clarity, ensuring that readers, especially those unfamiliar with specific terminology, can easily understand and follow the findings presented in the study.

2. Validation of Additional Genes in Fig. 6: I would suggest expanding the validation process to include a broader selection of differentially expressed genes (DEGs) along with immune related genes. This would provide a more comprehensive evaluation of the RNA-seq results and strengthen the overall findings.

Reviewer #3: Title

The title of the work is good

Abstract

The background is too lengthy. It would be better to state the results obtained with little explanation. In other words, the abstract should describe only the results obtained.

Materials and Methods

The word 'Infectious Model' would have been more preferred to 'Stimulation Model' despite the fact that it is an in vitro model The technique for the isolation and confirmation of the S. aureus strain used was not stated.

Results

The acronyms in table 1 should be explained. If there are replicates, the use of mean and standard deviation should have been ideal instead of the raw data.

The cluster analysis in Figure 2 has blurry explanation to the right of the diagram. The use of a more clearer diagram is necessary.

The number of novel genes in Table S2 are 40 but the reported genes in the result section is 39, what could be responsible for this discrepancy?

There should be consistency with 'Figure' and 'Fig'.

The work need to explain what the PPI represents (Figure 5).

Figure 6 has not been adequately explained. Reference should be made to what gene is upregulated or downregulated as the case may be.

Discussion

This section needs some additional adjustment. In this section the authors put the work in proper perspective in paragraph 1. However, subsequent paragraph that should be dedicated stating, explaining, interpreting and comparing the results with other studies, deliberated more effort in comparing the study with other works, thereby losing its intended content.

While it is understandable that epithelial cell is the first line of defense of a host, the authors appear to have deviated from the original intent of the work which is 'studying the transcriptome of ovine mammary epithelial cells stimulated with Staphylococcus aureus'. The authors now delve into mainly the immunological aspect.

Conclusion

This section need to be tailored towards the results obtained. It is ideal to be more specific as only one strain of the microorganism was used.

**Do you want your identity to be public for this peer review?** For information about this choice, including consent withdrawal, please see our Privacy Policy

Reviewer #1: No

Reviewer #2: **Yes: ** Arun Kumar Paripati

Reviewer #3: **Yes: ** Tombari Pius Monsi

---

## [Author Response · Author response to Decision Letter 1]

2 May 2025

Point-by-point response

High throughput transcriptomics analysis of ovine mammary epithelial cells stimulated with Staphylococcus aureus in vitro

Comment 1: 1. Thank you for including the following ethics statement on the submission details page:

'Experimental procedures used in this study were performed according to the Turkish council on animal experiment guidelines on farm animal facilities (15 February 2014, #28914).'

Please clarify the full name of the ethics committee that approved your animal study.

Response 1: Statement was added into Animal selection and sampling part “Experimental proce-dures used in this study were performed according to Institutional Animal Care and Use Committee of Erciyes University, Kayseri, Türkiye and research protocol adhered to the Turkish Council on Animal Experiment guidelines on farm animal facilities (15 February 2014, #28914).”

Comment 2: 2. Thank you for stating the following financial disclosure: “This project was sup-ported by Erciyes University Scientific Research Projects Unit under the code of FDK-2021-11437 and FYL-2023-12822.". Please state what role the funders took in the study. If the funders had no role, please state: "The funders had no role in study design, data collection and analysis, decision to publish, or preparation of the manuscript." If this statement is not correct you must amend it as needed. Please include this amended Role of Funder statement in your cover letter.

Response 2: The statement was added into the revised manuscript.

Comment 3: 3. Thank you for updating your data availability statement. You note that your data are available within the Supporting Information files, but no such files have been included with your submission. At this time we ask that you please upload your minimal data set as a Supporting Information file, or to a public repository such as Figshare or Dryad. Please also ensure that when you upload your file you include separate captions for your supplementary files at the end of your manuscript. As soon as you confirm the location of the data underlying your findings, we will be able to proceed with the review of your submission.

Response 3: The data that support the findings of this study are available in the Gene Expression Omnibus (GEO) [https://www.ncbi.nlm.nih.gov/geo/]. Accession number: GSE295455.

Reviewer #1

Dear authors, understanding the complex mechanisms of immune response against patho-gens is of particular importance. Your manuscript deals with the infection of ovine mammary epi-thelial cells with S. aureus.

The manuscript is nicely structured; however, a lot of information is missing to get a clear picture of the whole experiment. Further, some sections are described superficially, especially in the method and result part.

Comment No 1: Line 87: More information is needed on the ewes which were slaughter. Please add information related to the type of breed, the age of the animals and the number of slaughtered animals.

Response No 1: Thank you for your suggestion. We have added the details of breed (Akaraman), the age of the ewes ranging from 2-3 years and 3 animals were selected for sampling before slaughtering.

Comment No 2: Line 102/103: You collected samples from different ewes, where these samples pooled before establishing the cell culture model?

Response No 2: To maintain the identical cell culture conditions, yes samples were pooled before establishing stimulation model. It is common practice to minimize inter-individual genetic & physi-ological variations and to enhance the statistical power [1,2]

1. Takele Assefa, A., Vandesompele, J. and Thas, O., 2020. On the utility of RNA sample pooling to optimize cost and statistical power in RNA sequencing experiments. BMC ge-nomics, 21, pp.1-14.

2. Rajkumar, A.P., Qvist, P., Lazarus, R., Lescai, F., Ju, J., Nyegaard, M., Mors, O., Børglum, A.D., Li, Q. and Christensen, J.H., 2015. Experimental validation of methods for differen-tial gene expression analysis and sample pooling in RNA-seq. BMC genomics, 16, pp.1-8.

Comment No 3: How many replications were performed in the cell culture experiment? I have to look into the results section to get this information. Please provide the number here as well.

Response No 3: The primary cell culture consisted of a total of 12 replicates, with 4 replicates de-rived from each ewe sample. Prior to initiating the stimulation experiments, all samples were pooled to develop a comprehensive stimulation model.

Comment No 4: Line 106: Please provide a reference for EpCAM.

Response No 4: A reference for EpCAM has been included and highlighted in the revised manu-script [3].

3. Bach, K., Pensa, S., Grzelak, M., Hadfield, J., Adams, D. J., Marioni, J. C., & Khaled, W. T. (2017). Differentiation dynamics of mammary epithelial cells revealed by single-cell RNA sequencing. Nature communications, 8(1), 1-11.

Comment No 5: Line 145 to 147: You describe the thresholds for genes to be significantly differ-entially expressed. The problem of multiple tests (the importance of false discovery) is missing. In the result section (192 to 194) a method is described to control the false positive results. This meth-od needs to be explained in the method section as well, and which type this method controls false positive. Please also provide further information on the applied method.

Response No 5: Shrinking estimator using an empirical Bayes approach is explained and high-lighted in the Method section.

Comment No 6: Line 164: Which samples have been used here?

Response No 6: The same total RNA samples used for RNA sequencing were used here. We have clarified this in the revised text and highlighted it.

Comment No 7: As mentioned above, the result section is superficial and needs to be rewritten. Figure 5 and S2 are not described in the text or referenced. The results are presented in tables or figures, but they are not described in the text. It is unclear how to select 6 genes for the validation step.

Response No 7: The results section has been expanded with a detailed description of the figures, tables, and selection criteria for the ten (10) validated genes was based on their relevance to im-mune response, oxidative stress and statistical significance. These selection criteria are added in the revised manuscript and highlighted.

Comment No 8: Line 186: “there was a higher percentage of duplicated ….” Compared to which group? Please be more specific.

Response No 8: The proportion of duplicate reads across all sequenced datasets ranged between 91% and 93% in paired-end reads, indicating a high level of redundancy within the sequencing data. Sentence is revised, clarified and highlighted in revised manuscript.

It is important to mention here that we performed the analysis while keeping and removing dupli-cates, but duplicates caused the overestimation of transcript abundance, artificially inflating the expression of transcripts. Overestimated transcript counts can lead to false positives or exaggerated fold changes in differential expression analysis. As highlighted in studies by Baldoni et al. (2024) and Yuan et al. (2024), overestimated transcript counts due to PCR duplication distort the interpre-tation of differentially expressed genes (DEGs), leading to a higher false discovery rate (FDR). So, we removed the duplicates to reduce the bias, increase the accuracy of differential expression anal-ysis and prevent the false interpretation of biological significance. Principal component analysis also showed that the read counts reflect true biological variation. Furthermore, we validate the se-lected DEGs using RT-qPCR techniques.

Yuan, Y., Xu, Q., Wani, A., Dahrendorff, J., Wang, C., Shen, A., ... & Qu, A. (2024). Differen-tially expressed heterogeneous overdispersion genes testing for count data. Plos one, 19(7), e0300565.

Baldoni, P. L., Chen, L., & Smyth, G. K. (2024). Faster and more accurate assessment of dif-ferential transcript expression with Gibbs sampling and edgeR 4.0. bioRxiv, 2024-06.

Comment No 9: Line 189: In table 1 you see differences between the raw reads of the control group and the stimulated group? Please give an explanation.

Response No 9: The differences in raw reads between the control and stimulated groups may arise due to biological variations in response to stimulation, as well as technical factors such as sequenc-ing efficiency and RNA extraction variability. To ensure that these differences do not introduce bias in downstream analyses, we applied DESeq2’s variance stabilizing transformation (VST) for normalization. This method effectively corrects differences in sequencing depth and variability, allowing for accurate comparison of gene expression levels between groups. By using VST, we minimized the impact of raw read differences and ensured that downstream differential expression analysis was reliable. This clarification is included and highlighted in the revised manuscript.

Comment No 10: Line 227 to 230: Why were some GO groups overrepresented? How did you observe this from the figures given?

Response No 10: Overrepresentation is represented by the relative sizes of the GO categories in the given figures. Larger blocks indicate a higher frequency of associated genes within these cate-gories. It is clarified and highlighted in description of figures of revised manuscript.

Comment No 11: The discussion mainly focused on the functional description of the genes, be-cause there is a lack of comparable studies. An outlook on further research is missing. The listed experiments were performed in other species. Could you provide additional information on the experiment design in order to classify the results obtained here?

Response No 11:

Reviewer #2:

In the current manuscript, authors conducted a transcriptomic study to identify host defense genes in ovine mammary epithelial cells challenged with a field strain of Staphylococcus aureus. Overall, the study and manuscript appear satisfactory, though I have a few minor concerns.

Comment No 1: Inclusion of Gene Abbreviations: It would be beneficial for the authors to include the full names or abbreviations for all the genes mentioned throughout the manuscript. This would improve clarity, ensuring that readers, especially those unfamiliar with specific terminology, can easily understand and follow the findings presented in the study.

Response No 1: We have revised the manuscript to ensure the gene abbreviations, full names were included in Table S2.

Comment No 2: Validation of Additional Genes in Fig. 6: I would suggest expanding the valida-tion process to include a broader selection of differentially expressed genes (DEGs) along with immune related genes. This would provide a more comprehensive evaluation of the RNA-seq re-sults and strengthen the overall findings.

Response No 2: We have included additional genes in the validation process to strengthen the findings.

Reviewer #3

Title

Comment No 1: The title of the work is good

Response No 1: Thank you for your appreciation.

Abstract

Comment No 2: The background is too lengthy. It would be better to state the results obtained with little explanation. In other words, the abstract should describe only the results obtained.

Response No 2: The abstract is revised and improved according to reviewer’s suggestions. The background was shortened, and explanation of the results was added and highlighted in revised manuscript.

Materials and Methods

Comment No 3: The word 'Infectious Model' would have been preferred to 'Stimulation Model' despite the fact that it is an in vitro model. The technique for the isolation and confirmation of the S. aureus strain used was not stated.

Response No 3: We have changed 'Stimulation Model' to 'Infectious Model' and included details on the isolation and confirmation of the S. aureus strain as follows:

Milk samples were plated on blood agar plates (Oxoid, UK) supplemented with 5% defibrinated ovine blood and incubated at 37 °C for 24 to 48 hours. Following incubation, suspected bacterial colonies were subjected to Gram staining and examined under a light microscope. Gram-positive cocci were further characterized using a series of biochemical tests. The biochemical properties of the isolates were assessed through catalase activity, hemolysis on blood agar, coagulase activity, nitrate reduction, DNase agar, clumping factor presence, arginine dihydrolase activity, and urease production, as described by Quinn et al. (1998). Coagulase-positive isolates were further analyzed using the API STAPH IDENT system, 32 Staph (bioMérieux SA, 69280 Marcy-l'Étoile, France), to confirm the identification of S. aureus.

Results

Comment No 4: The acronyms in table 1 should be explained. If there are replicates, the use of mean and standard deviation should have been ideal instead of the raw data.

Response No 4: We have provided explanations for acronyms in the revised version. All raw and clean data statistics are provided for all replicates (3 control × 3 stimulated)

Comment No 5: The cluster analysis in Figure 2 has blurry explanation to the right of the diagram. The use of a clearer diagram is necessary.

Response No 5: The right side of the diagram displays gene names corresponding to each cluster. The cluster analysis effectively illustrates the variability in normalized gene expression levels across different samples and replicates, capturing this variability comprehensively. To address the concern raised, we have replaced the original figure with an updated, clearer version to improve visual interpretation and understanding.

Comment No 6: The number of novel genes in Table S2 are 40 but the reported genes in the result section is 39, what could be responsible for this discrepancy?

Response No 6: This discrepancy was due to a typographical error, which has been corrected and highlighted in revised manuscript.

Comment No 7: There should be consistency with 'Figure' and 'Fig'.

Response No 7: We have ensured consistency throughout the revised manuscript.

Comment No 8: The work needs to explain what the PPI represents (Figure 5).

Response No 8: We have added and highlighted a detailed explanation of PPI network in revised manuscript under the section ‘Protein-protein interaction (PPI) analysis of DEGs’ in results.

Comment No 9: Figure 6 has not been adequately explained. Reference should be made to what gene is upregulated or downregulated as the case may be.

Response No 9: In the RT-qPCR validation, 7 upregulated and 3 downregulated DEGs were se-lected based on relevance to study and statistical significance. It was clearly described and high-lighted in the revised manuscript.

Discussion

Comment No 10: This section needs some additional adjustment. In this section the authors put the work in proper perspective in paragraph 1. However, subsequent paragraph that should be ded-icated stating, explaining, interpreting and comparing the results with other studies, deliberated more effort in comparing the study with other works, thereby losing its intended content.

While it is understandable that epithelial cells are the first line of defense of a host, the authors ap-pear to have deviated from the original intent of the work which is 'studying the transcriptome of ovine mammary epithelial cells stimulated with Staphylococcus aureus'. The authors now delve into mainly the immunological aspect.

Response No 10: An additional adjustment was made by adding one more paragraph with ex-plaining, interpreting and comparing the results beyond the immunological aspect. Highlighted in revised manuscript.

Conclusion

Comment No 11: This section needs to be tailored towards the results obtained. It is ideal to be more specific as only one strain of the microorganism was used.

Response No 11: We have made the conclusion more specific to the study findings and highlight-ed them in revised manuscript, emphasizing that OMECs response to only one causative agent; S. aureus was analyzed in this transcriptomic study.

---

## [Decision Letter · Decision Letter 1]

3 Aug 2025

Dear Dr. Cinar,

Thank you for submitting your manuscript to PLOS ONE. After careful consideration, we feel that it has merit but does not fully meet PLOS ONE’s publication criteria as it currently stands. Therefore, we invite you to submit a revised version of the manuscript that addresses the points raised during the review process.

We look forward to receiving your revised manuscript.

Kind regards,

Muhammad Ahmad

Academic Editor

PLOS ONE

Journal Requirements:

Additional Editor Comments:

Dear Author,

The reviewers' feedback has been generally positive, and your manuscript is moving toward acceptance pending revisions. Although the reviewers have accepted the core content, I have come across some relevant and interesting papers that could further enhance the quality and depth of your work.

http://www.ijvets.com/pdf-files/23-198.pdf

https://agrobiologicalrecords.com/detail.php?view_id=2188

https://vetdergikafkas.org/uploads/pdf/pdf_KVFD_3172.pdf

https://www.pvj.com.pk/pdf-files/25-149.pdf

I recommend reviewing these papers and incorporating relevant information or citations where appropriate to strengthen the manuscript.

Thank you for your continued efforts.

Reviewers' comments:

Reviewer's Responses to Questions

**Comments to the Author**

Reviewer #2: All comments have been addressed

Reviewer #4: (No Response)

2. Is the manuscript technically sound, and do the data support the conclusions?

Reviewer #2: Yes

Reviewer #4: Yes

3. Has the statistical analysis been performed appropriately and rigorously?

Reviewer #2: Yes

Reviewer #4: Yes

4. Have the authors made all data underlying the findings in their manuscript fully available?

Reviewer #2: Yes

Reviewer #4: Yes

5. Is the manuscript presented in an intelligible fashion and written in standard English?

Reviewer #2: Yes

Reviewer #4: Yes

Reviewer #2: (No Response)

Reviewer #4: The article is well-structured, clearly written, and presents a solid dataset supporting its conclusions. The section discussing differential gene expression (DEG) analysis and the immune response in ovine mammary epithelial cells (oMECs) following Staphylococcus aureus stimulation is generally well explained and informative. However, a minor revision is recommended to improve the clarity and scientific rigor of the Discussion section.

1. Line 271: The phrase "|log2 (fold change)| < 1" seems inconsistent with standard DEG filtering, as typically DEGs are considered significant if the absolute log2 fold change is greater than or equal to 1. If this is a typo, it should be corrected to "|log2 (fold change)| ≥ 1". Otherwise, the rationale for using a <1 threshold should be justified.

2. Line 272: Strengthen transitions between comparative studies. The comparisons with other studies (e.g., Chen et al., [1]) are useful but could benefit from more concise phrasing and a clearer connection to the current study's unique findings. Please see “Riaz, M. et al. (2025). Transcriptomic insights into the bovine immune response following Staphylococcus aureus infection. International Journal of Veterinary Science, 14(3), 200–208. Available at: https://www.ijvets.com/article/264/” as a suggested citation and a supporting reference to strengthen your argument.

3. While the discussion on high duplicate reads is relevant, it could be summarized succinctly or referenced with greater clarity regarding how it affects DEG reliability in this study.

4. The interpretation of upregulated genes such as RPS19, hnRNPs, and ACTB is commendable. Still, it could be further refined by linking their immune roles more directly to the context of S. aureus infection in oMECs. The manuscript would benefit from a deeper exploration of the immune pathways' molecular intricacies.

**Do you want your identity to be public for this peer review?** For information about this choice, including consent withdrawal, please see our Privacy Policy

Reviewer #2: **Yes: ** Arun Kumar Paripati

Reviewer #4: No

---

## [Author Response · Author response to Decision Letter 2]

5 Sep 2025

Point-by-point response

High throughput transcriptomics analysis of ovine mammary epithelial cells stimulated with Staphylococcus aureus in vitro

Academic editor

Comment: The reviewers' feedback has been generally positive, and your manuscript is moving toward acceptance pending revisions. Although the reviewers have accepted the core content, I have come across some relevant and interesting papers that could further enhance the quality and depth of your work. I recommend reviewing these papers and incorporating relevant information or citations where appropriate to strengthen the manuscript.

Response:

Thank you for your constructive suggestions and for highlighting additional literature relevant to our study. We have carefully reviewed the papers you provided. The study by Nadi et al. (2023) examines the prevalence of Staphylococcus aureus, Pseudomonas aeruginosa and Escherichia coli in dairy products and reports a high frequency of methicillin resistance among Staphylococcus aureus isolates; this citation has been incorporated into the introduction section of manuscript to emphasize the public health significance of antimicrobial‑resistant Staphylococcus aureus strains. The work by Torun et al. (2025) compares virulence and resistance genes among coagulase‑negative staphylococci and Staphylococcus aureus from raw milk, we now reference this study in the introduction to underscore the diversity of staphylococcal species and the prevalence of the mecA gene. Zafar et al. (2025) demonstrate that Staphylococcus aureus is the most frequent cause of mastitis and explore the antibacterial potential of copper nanoparticles. We cite this study to highlight the alternative therapeutic strategies against mastitis. These citations strengthen the context of our research and underscore the relevance of our transcriptomic analysis. The recommended reference; Rashid et al. (2024) was not cited, as it was not relevant to the present study.

Reviewer #4

Comment: 1. The article is well-structured, clearly written, and presents a solid dataset supporting its conclusions. The section discussing differential gene expression (DEG) analysis and the immune response in ovine mammary epithelial cells (oMECs) following Staphylococcus aureus stimulation is generally well explained and informative. However, a minor revision is recommended to improve the clarity and scientific rigor of the Discussion section.

Response: Thank you for acknowledging our work and for your positive feedback regarding the structure, clarity and data set presented in the manuscript. We appreciate your constructive suggestion to enhance the clarity and scientific rigor of the Discussion section. In response, we have carefully revised this section to better contextualize our findings.

Comment No 1: 1. Line 271: The phrase "|log2 (fold change)| < 1" seems inconsistent with standard DEG filtering, as typically DEGs are considered significant if the absolute log2 fold change is greater than or equal to 1. If this is a typo, it should be corrected to "|log2 (fold change)| ≥ 1". Otherwise, the rationale for using a <1 threshold should be justified.

Response No 1: We thank the reviewer for pointing out this inconsistency. The phrase "|log₂ (fold change)| < 1" was indeed a typographical error. As correctly noted, the standard threshold for identifying differentially expressed genes is "|log₂ (fold change)| ≥ 1". We have reviewed the manuscript thoroughly and corrected this error to ensure consistency throughout the text

Comment No 2: 2. Line 272: Strengthen transitions between comparative studies. The comparisons with other studies (e.g., Chen et al., [1]) are useful but could benefit from more concise phrasing and a clearer connection to the current study's unique findings. Please see “Riaz, M. et al. (2025). Transcriptomic insights into the bovine immune response following Staphylococcus aureus infection. International Journal of Veterinary Science, 14(3), 200–208. Available at: https://www.ijvets.com/article/264/” as a suggested citation and a supporting reference to strengthen your argument.

Response 2: Thank you for noting that the comparative section of the discussion could be more concise. In the revised manuscript we extended the section comparing our findings with previous studies. Although the recommended study was not accessible because of incorrect information International Journal of Veterinary Science, volume 14(3) contained page 434-623 (https://www.ijvets.com/volume-14-no-3-2025/ ). Even volume 14(2) and 14(1) does not contain any such article (https://www.ijvets.com/volume-14-no-2-2025/,
https://www.ijvets.com/volume-14-no-1-2025/ ).

Comment No 3: 3. While the discussion on high duplicate reads is relevant, it could be summarized succinctly or referenced with greater clarity regarding how it affects DEG reliability in this study.

Response No 3: We appreciate your suggestion to summarize the discussion of duplicate reads. We have condensed this part and clarified that the high proportion of duplicate reads in our dataset likely reflects the saturation of mapping space by highly expressed transcripts rather than technical artefacts. We also refer to existing literature to explain that removal of duplicates can bias expression estimates when duplicates originate from abundant transcripts.

Comment No 4: 4. The interpretation of upregulated genes such as RPS19, hnRNPs, and ACTB is commendable. Still, it could be further refined by linking their immune roles more directly to the context of S. aureus infection in oMECs. The manuscript would benefit from a deeper exploration of the immune pathways' molecular intricacies.

Response No 4: Thank you for encouraging a deeper exploration of the immune roles of the up‑regulated genes. We expanded the discussion to link RPS19, RPS14, hnRNPs and ACTB to specific innate immune pathways. For example, we note that RPS19 interacts with macrophage migration inhibitory factor, ERK and NF‑κB, bridging translation and inflammatory signalling; RPS14 contributes to TLR4 recognition of Staphylococcus aureus. The hnRNP family members regulate the stability of transcripts encoding cytokines such as TNF‑α and IL‑6. Similarly, ACTB participates in cytoskeletal rearrangements required for pathogen internalisation and leukocyte migration. We also discuss how these genes fit into broader pathways such as TLR, NOD‑like receptor, NF‑κB and MAPK signalling, thereby providing a more comprehensive interpretation of our transcriptomic data.

---

## [Editor Report · Decision Letter 2]

15 Sep 2025

High-throughput transcriptomics analysis of ovine mammary epithelial cells stimulated with Staphylococcus aureus in vitro

PONE-D-25-01127R2

Dear Dr. Cinar,

We’re pleased to inform you that your manuscript has been judged scientifically suitable for publication and will be formally accepted for publication once it meets all outstanding technical requirements.

Kind regards,

Muhammad Ahmad

Academic Editor

PLOS ONE
---

## [Editor Report · Acceptance letter]

PONE-D-25-01127R2

PLOS ONE

Dear Dr. Cinar,

I'm pleased to inform you that your manuscript has been deemed suitable for publication in PLOS ONE. Congratulations! Your manuscript is now being handed over to our production team.

Kind regards,

on behalf of

Mr. Muhammad Ahmad

Academic Editor

PLOS ONE